# The supply is there. So why can't pregnant and breastfeeding women in rural India get the COVID-19 vaccine?

Nadia G. Diamond-Smith[1]*, Preetika Sharma[2], Mona Duggal[2], Navneet Gill[2], Jagriti Gupta[3], Vijay Kumar[3], Jasmeet Kaur[4], Pushpendra Singh[4], Katy Bradford Vosburg[1], Alison M. El Ayadi[1]

1 University of California, San Francisco, CA, United States of America, 2 Post Graduate Institute of Medical Education and Research, Chandigarh, India, 3 Survival of Women and Children Foundation, Panchkula, India, 4 Department of Computer Science & Engineering, Indraprastha Institute of Information Technology Delhi, New Delhi, India

* nadia.diamond-smith@ucsf.edu

**Data Availability Statement:** DOI: Diamond-Smith, Nadia et al. (2022), The supply is there. So why can't pregnant and breastfeeding women in rural

## Abstract

Despite COVID-19 vaccines being available to pregnant women in India since summer 2021, little is known about vaccine uptake among this high need population. We conducted mixed methods research with pregnant and recently delivered rural women in northern India, consisting of 300 phone surveys and 15 in-depth interviews, in November 2021. Only about a third of respondents were vaccinated, however, about half of unvaccinated respondents reported that they would get vaccinated now if they could. Fears of harm to the unborn baby or young infant were common (22% of unvaccinated women). However, among unvaccinated women who wanted to get vaccinated, the most common barrier reported was that their health care provider refused to provide them the vaccine. Gender barriers and social norms also played a role, with family members restricting women's access. Trust in the health system was high, however, women were most often getting information about COVID-19 vaccines from sources that they did not trust, and they knew they were getting potentially poor-quality information. Qualitative data shed light on the barriers women faced from their family and health care providers but described how as more people got the vaccine that norms were changing. These findings highlight how pregnant women in India have lower vaccination rates than the general population, and while vaccine hesitancy does play a role, structural barriers from the health care system also limit access to vaccines. Interventions must be developed that target household decision-makers and health providers at the community level, and that take advantage of the trust that rural women already have in their health care providers and the government. It is essential to think beyond vaccine hesitancy and think at the system level when addressing this missed opportunity to vaccinate high risk pregnant women in this setting.

India get the COVID-19 vaccine? , Dryad, Dataset,
https://doi.org/10.7272/Q6XD0ZX8.

**Funding:** This project was funded by the Vaccine
Confidence Fund, Grant ID VCF – 028 (Co-PI AEA
and ND-S). The funders had no role in study
design, data collection and analysis, decision to
publish, or preparation of the manuscript.

**Competing interests:** The authors have declared
that no competing interests exist.

## Introduction

As of December 2021, the Ministry of Health India reported that 61% of the adult population
is fully vaccinated, almost reaching the WHO target for vaccination coverage for all countries
to reach 70% by the end of 2022 (Fig 1) [1]. Global inequities in COVID-19 vaccine distribu-
tion remain significant; however, vaccine availability is considered robust in most high and
many middle-income countries, including India—a major producer of vaccines.

Despite these overall high rates in India, there are significant disparities in vaccination rates
across income, region, education, age and especially gender [1]. Several studies highlight that
individuals with an advanced education and job title are more likely to be aware of the
COVID-19 vaccine and accepting of the vaccine [2, 3]. In India gender has played an impor-
tant role in structuring how the COVID-19 pandemic has impacted health outcomes with
women bearing an unequal COVID-19 mortality compared to men, including among younger
women of reproductive age [4, 5]. This is in contrast to trends globally, where men had higher
mortality than women [6]. Gender inequality is especially important to consider in India,
where strict, inequitable gender norms often limit women's autonomy, decision-making
power, mobility, access to care, and overall levels of empowerment.

COVID-19 infection during pregnancy is of particular concern as pregnant women are
more likely to develop severe disease from COVID-19 infection. COVID-19 during pregnancy
is also associated with elevated risks of fetal and neonatal complications [7, 8]. Unvaccinated
pregnant women are at even more risk of having adverse birth outcomes [9]. COVID-19 con-
tributed greatly to maternal mortality as well. Shortly before the Indian government issued
guidance that the COVID-19 vaccine was approved for pregnant and breastfeeding women in
India (July 2nd, 2021, Fig 1, before our study took place), Mumbai, the second most populous
city in India, declared that COVID-19 was the leading cause of death among new mothers

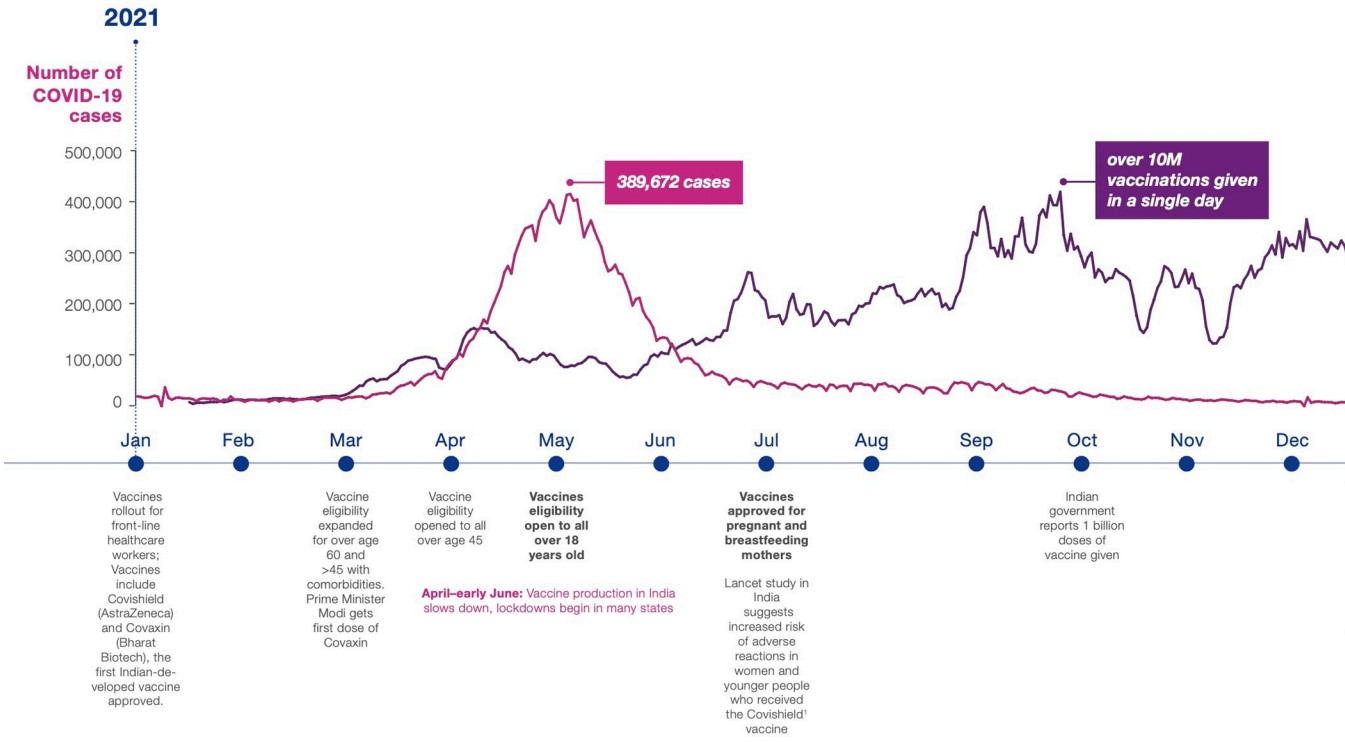

**Fig 1. 2021 Timeline of vaccine distribution and restrictions in India.**

between 2020–2021 [10]. However, despite vaccine availability for Indian pregnant women in the six months between then and when this data was collected, little is known about current uptake in this population. Currently, CoWin, which tracks India's COVID-19 vaccine statistics, does not have the ability to look at vaccination rates broken down by key characteristics such as for pregnant or postpartum women, socio-economic status, tribal communities, or other often overlooked subpopulations. A multi-country study conducted before the vaccine was approved for pregnant women and found that most pregnant Indian respondents (93%) reported they were likely to get the COVID-19 vaccine, a substantially higher proportion than other countries included in the study [11]. However, we know little about how this translated into uptake among pregnant women once the vaccine was approved.

A 2021 systematic review of studies looking at vaccine uptake in pregnant women found that the biggest predictors of vaccine uptake included trust in the effectiveness and safety of the vaccine, communication about safety of the vaccine in pregnancy, trust in public health agencies, anxiety about COVID-19 and other COVID-19 safety behaviors [12]. Age, education and socio-economic status were also associated with vaccination in pregnant women. However, this review mostly included papers with data from high income countries, with only one paper from 2020 having collected data in India. While there are a number of papers from across India looking at vaccine acceptance and hesitancy [2, 13–16], only one focuses on pregnant and postpartum women. The 2022 study from northern India explored barriers to vaccination among pregnant and postpartum women, collecting data from a mostly urban, well-educated and highly vaccinated population. This study found large differences by socio-economic and rural/urban status, and found that concerns over the impacts on the baby and lack of inclusion of women in vaccine safety trials were main contributors to vaccine hesitancy (delay in acceptance or refusal of vaccination despite availability) [17]. More, mixed-methods research is needed with rural, less well off, and un-vaccinated groups of pregnant and postpartum women to understand contributors and barriers to vaccine uptake. With this in mind, we hypothesized that a variety of factors would influence vaccine uptake among pregnant and recently delivered women in India. Specifically, belief in the effectiveness and safety, communication from trusted health care providers, norms around community acceptance/uptake, and socio-economic factors.

## Materials and methods

### Ethics statement

This study received ethical approval from the Human Research Protection Program at the University of California, San Francisco (#21–35278) and Indraprastha Institute of Information Technology, Delhi Institutional Review Board (IIITD/IRB/07/2021/03). All participants underwent an informed consent process and confirmation was obtained verbally.

To understand vaccine uptake, barriers, hesitancy, facilitating factors and sources of trusted information among pregnant and breastfeeding women, we conducted mixed-methods research in northern India in November 2021. Broadly, the study was carried out in lower and upper middle class populations. The WhatsApp groups through which we recruited (described below) was designed to serve a population residing in rural and suburban areas in two districts in Haryana. However, since women migrate, our estimates are that about 60% of the respondents were from these two districts in north India, the rest were most likely from Punjab, UT Chandigarh, Uttar Pradesh, Uttarakhand or UT Delhi. In terms of health and socio demographic characteristics, these districts rank in the middle, compared to other Indian districts, during the past decade. In this setting, the vaccine was available free of cost in government facilities, and for a charge in private facilities. In total, we conducted 300 phone surveys and 15

in-depth interviews with women. The eligibility criteria were to include pregnant and recently delivered women who were breastfeeding (up to one year postpartum). Since this was a formative, descriptive study, we estimated that 300 respondents would be sufficient to describe trends in vaccination, barriers and hesitation.

The surveys were conducted telephonically. The participants were active members of WhatsApp groups run by a local NGO that was a collaborator on the project. All women in the WhatsApp group were connected to the government health care system, which provides free services. A list of 552 eligible women, from a sample of about 5,000, was provided to the research assistants. Women who were either pregnant or had delivered within 1 year were eligible for the survey. The list included their name, mobile and date of delivery. These women were called one by one down the list provided by the research assistant. Women were read an informed consent and asked to provide verbal consent. A survey call was scheduled based on time convenient for the women. Most of the surveys were completed in one call and few were done in parts based on the availability of the participant. Out of about 450 women called, 300 complete surveys were taken. Some women did not pick the call, or only completed half of the survey. The team began to take the surveys in the first week of November, 2021 and 300 surveys were completed by Nov 27, 2021. The women were followed up with to get the missing information wherever possible.

The survey tool was developed and finalized by the team members and was put on web application RedCap. It included questions on vaccine acceptance, barriers, hesitancy and socio-demographics. The redcap data underwent quality assessments and then incorrect data or missed areas were addressed. Quantitative data associated with this study are publicly accessible in Dryad [18].

Quantitative survey data was analyzed using descriptive statistics (frequencies, percentages) using STATA version 15 [19]. We then explored sets of factors associated with being vaccinated (vaccinate yes/no) using logistic regression models. Our primary predictor of interest was pregnancy status, coded as currently pregnant or postpartum. First, we looked as sociodemographic factors, including age (continuous), schooling (Some secondary or more compared (to Primary or less), religion (Hindu compared to other religion), caste (General caste compared to other caste), and having a ration card or not, which is a marker of poverty status. Ration cards are provided based on the income level. Next, we looked at three measures associated with confidence in the vaccine (all coded "not at all, "somewhat", "moderately" or "very", but changed into a binary of "very" vs. others), including believing that the vaccine is effective, safe for the woman and safe for the baby. Next, we looked at two questions associated with trust in the medical system, (with the same four answer options as above, any made into a binary) including how much the woman trust the public health agencies that recommend the vaccine and how much they trust their perinatal care providers in recommending the vaccine. The next model looked specifically at if their perinatal care provider had recommended that they get the vaccine, told them NOT to get it while pregnant, or had not mentioned it. Next, we looked at the impact of information using a question which asked if the respondent had heard or seen any information about COVID-19 vaccines that she could not tell if was true or false (yes/no or not sure). Finally, we looked at two measures of social pressure, including whether the respondent thought that most of their community would get the vaccine (yes/no) and most of their family would get the vaccine (yes/no).

## In depth interviews

At the end of the telephonic survey, women were asked if they would be willing to be contacted to participate in longer follow-up in-depth interviews. Later, the research associate reached out

to those who agreed for a telephonic in-depth interviews. Of the 27 women called, 15 women agreed to give an interview. In depth interviews lasted from 35–70 minutes. In depth interviews were collected until data saturation was reached, which was determined by the study team conducting the interviews. Some surveys and interviews were taken on multiple calls due to the time constraints of the participants. The interviews were conducted in Hindi and were later transcribed and translated into English. A codebook based on the key themes covered in the interview was made to analyze the interviews, which was done by the three members from the research team (XX XX XX) using Dedoose software (Version 8). In the context of the COVID-19 vaccine, key themes covered in the interview included general impact of COVID-19 pandemic, its impact on their pregnancy, their own thoughts on COVID-19 vaccine, thoughts of their family members, views of community and perinatal care provider on COVID-19 vaccine, decision making, sources of information and understanding of technology.

## Results

The mean age of respondents from the surveys was 26, ranging from 17–42 (Table 1). Just under half of respondents had secondary education or more (46%), most were Hindu (92%), a third were general caste (66.7%), and just over half had a ration card (56%). Respondents from the qualitative interviews were similar: mean age was 26, just under half had secondary education (46%), 60% were general caste, however, fewer had a ration card (33%).

Vaccination rates were low; but desire for vaccination was high. Only one-third (36%) of pregnant and breastfeeding women had received the vaccine (Table 2). This is despite all women registering their pregnancies with either a public or private health facility, indicating that they had seen a health care provider during their pregnancy—a potential point of contact for vaccination. All of our respondents were engaged in the health system and regularly receiving care throughout their pregnancies, and all of their pregnancies occurred after vaccines were approved for pregnant women. About half (47%) of non-vaccinated women said they would get the vaccine now if they could, an additional 20% want to get it soon but would wait, and 27% reported that they would not get it at all (the remaining were unsure). As one non-vaccinated women explained:

> *"I have not got vaccinated so I can't say but if you will ask me then I think it will have a good effect on the baby. Like there will be some immunity power that will be generated due to the Covid-19 vaccine. Just like we eat everything, that gives the baby the nutrition and the same way as my baby had the growth even though I took the injections." (age 30, breastfeeding)*

**Table 1. Demographics of the population.**

| | Survey | | In-depth interviews | |
|---|---|---|---|---|
| | Number/mean | Percent /range | Number/mean | Percent /range |
| Age (mean, range) | 26 | (17–42) | 26 | (22–31) |
| Education | | | | |
| <secondary | 137 | 45.7% | 7 | 46% |
| Secondary or more | 163 | 54.3% | 8 | 54% |
| Religion (Hindu) | 275 | 91.7% | | |
| Caste (Other = schedule caste and other backward class, compared to general) | 200 | 66.7% | 9 | 60% |
| Has Ration Card | 164 | 54.7% | 5 | 33.33% |

**Table 2. Vaccination practices and beliefs.**

| | Have you received a COVID-19 vaccine? | | | |
|---|---|---|---|---|
| | No | | Yes | |
| | N = 191 | % | N = 103 | % |
| **Do you think the vaccine is effective?** | | | | |
| No | 13 | 6.8 | 3 | 2.8 |
| Yes | 178 | 93.2 | 106 | 97.2 |
| **Do you think the vaccine is safe for the you?** | | | | |
| No | 21 | 11 | 1 | 0.9 |
| Yes | 170 | 89 | 108 | 99.1 |
| **Do you think the vaccine is safe for the baby?** | | | | |
| No | 26 | 13.6 | 6 | 5.5 |
| Yes | 165 | 86.4 | 103 | 94.5 |
| **How much do you trust the public health agencies that recommend you get a COVID-19 vaccine?** | | | | |
| Not at all | 2 | 1 | 2 | 1.8 |
| A little | 1 | 0.5 | 0 | 0 |
| Moderately | 22 | 11.5 | 3 | 2.8 |
| Very much | 166 | 86.9 | 104 | 95.4 |
| **How much do you trust your perinatal care providers in recommending you get the COVID-19 vaccine?** | | | | |
| Not at all | 2 | 1.1 | 3 | 2.8 |
| A little | 2 | 1.1 | 0 | 0 |
| Moderately | 21 | 11.1 | 2 | 1.8 |
| Very much | 165 | 86.8 | 104 | 95.4 |
| **Have you seen or heard any information about COVID-19 vaccines?** | | | | |
| No | 83 | 43.5 | 33 | 30.3 |
| Yes | 108 | 56.5 | 76 | 69.7 |
| **Do you know where to get accurate, timely information about COVID-19 vaccine?** | | | | |
| Yes | 7 | 3.66 | 4 | 3.67 |
| No | 181 | 94.76 | 104 | 95.41 |
| Not sure | 3 | 1.57 | 1 | 0.92 |
| **Do you think that most of your friends and family will get a COVID-19 vaccine?** | | | | |
| No | 12 | 6.3 | 1 | 0.9 |
| Yes | 179 | 93.7 | 107 | 99.1 |
| **Do you think that most of the people in your community will get a COVID-19 vaccine?** | | | | |
| No | 12 | 6.3 | 0 | 0 |
| Yes | 179 | 93.7 | 109 | 100 |
| **Has your perinatal care provider recommended that you get the COVID-19 vaccine?** | | | | |
| Yes—recommended vaccine | 33 | 17.3 | 41 | 37.6 |
| No—recommended against vaccine | 89 | 46.6 | 21 | 19.3 |
| No—has not addressed vaccine | 69 | 36.1 | 47 | 43.1 |
| **Among those not vaccinated:** | | | | |
| **If a COVID-19 vaccine were available to you, would you get it?** | | | | |
| Yes, I would get it as soon as possible | 89 | 47.1% | | |
| Yes, but would wait to get it | 38 | 20.1% | | |
| No | 50 | 26.5% | | |
| Not sure | 12 | 6.3% | | |
| **How easy do you think it will be to get a COVID-19 vaccine for yourself?** | | | | |
| Very easy | 92 | 84.4% | | |
| Somewhat easy | 10 | 9.2% | | |

*(Continued)*

**Table 2.** (Continued)

| | Have you received a COVID-19 vaccine? | | | |
|---|---|---|---|---|
| | No | | Yes | |
| | N = 191 | % | N = 103 | % |
| Somewhat difficult | 2 | 1.8% | | |
| Very difficult | 5 | 4.6% | | |

Below we describe the main themes of facilitators and barriers women faced to getting the vaccine that they wanted:

## Vaccine safety

Most (95%) thought that the vaccine was effective and safe for the mother (93%) and baby (89%), these percentages were high among those who were vaccinated and those who were not (Table 2). Concerns related to health of the mother and baby were common in the survey data; for example, 22% were worried about safety for their baby in one way or another. Most (65%) had no concerns and 13% some other concerns (family members concern, general lack of trust, confusion, general "safety"); of note, none reported concerns regarding the mother's health. Only 6% of women who were vaccinated reported concerns over safety, while 26% of women who were not vaccinated did. In our qualitative research, women described their own, or their family member's worry about the vaccine spoiling breast-milk quality and quantity, and the potential for impact on the physical and mental development of child.

## Refusal on the part of health care providers

One of the next common reasons reported that women who wanted the vaccine had not gotten it was that community health workers and doctors were refusing to administer the vaccine to pregnant women; 13% of women who were not vaccinated reported this as the reason why. We did not specifically provide this response option in our questionnaire since we were not anticipating this situation; thus, we hypothesize this could have been the case for more women, however, we only have data on the women who selected "other" as a response option and told the interviewer that this was their reason. For example, women responded to the "other" option with statements like "I wanted to get it but dispensary staff and Nurse denied to vaccinate me. What should I do?" and "This vaccine is very effective & safe. Everyone should take this, but my doctor forbade me to take the vaccine due to my pregnancy." This theme also came up repeatedly in the qualitative interviews, again suggesting that this sentiment was even more common than survey data suggest. As women seeking COVID-19 vaccination described, they went to seek the vaccine, but were denied by health care providers—who they rely on for information:

> "*I went to hospital a number of times but they refused to vaccinate me by saying that you are pregnant so you can't have it. Come after delivery*." (age 30, breastfeeding)

> "*When I was pregnant, I went to ask they said that it is not allowed to get the vaccination during pregnancy and after delivery is permitted. I don't have much knowledge, who should take it and who shall not.*" (age 25, breastfeeding)

Even women in the postpartum period described being denied the vaccine "*When my child was 7 months old, I asked the ASHA, she said that it is not allowed to you and then I did not ask*" (age 23, breastfeeding)

This finding is especially concerning given our study population reported high levels of trust in the health care system, with 90% of participants saying they trusted the government/ public health agencies "very much" and 90% saying they trusted their perinatal care provider "very much". Again, more women who were vaccinated reported high levels of trust compared to unvaccinated women (Table 2). Yet, only 25% of respondents were recommended to receive the vaccine by their providers. One respondent noted: *"If Government has introduced a vaccination then it should be tried."* (age 28, breastfeeding). Where public or private healthcare providers recommended the vaccine, patients readily accepted it:

"*My mother had a word with the [community health worker] regarding COVID-19 vaccine. She suggested us to have it. Actually, we had discussed it with our doctor and they gave us approval only that's why we had vaccination otherwise we would not have taken it" (age 25, pregnant).*

Furthermore, even providers not discussing or mentioning the vaccine led women not to take it up, as one woman from the survey described "*No one has suggested me to take my vaccination. If it's necessary then I would have it.*"

## Gender and subjective norms

Aside from reluctance among health care providers to vaccinate pregnant women, the role of gender also contributed to low vaccine uptake among pregnant and breastfeeding women in our sample. As mentioned above, in the Indian setting, decisions are often made at the household level, by men or in-laws. Restrictions on mobility and norms around how women can interact with others in society were a barrier to information, as one respondent described:

"*I don't know much because I stay home and rarely go out. I don't talk to people in the community. My in-laws go out and they share what is happening in society.*" (age 24, breastfeeding).

Low empowerment made it difficult for some women to advocate for vaccination or to seek information they needed about the vaccine. In the qualitative interviews, one woman who did get vaccinated when her baby was an infant described how her mother-in-law would not let her hold her baby for many hours after her vaccination.

In addition to gender norms and barriers, we observed broader subjective norms influencing behavior. As one woman explained:

"*They [family members] were not agreeing for it [the COVID-19 vaccine] because others were not going to get it and now they are lining up in vaccination camps because they are seeing others' vaccination results [no side effects, helping people]" (age 24, pregnant)*

However, survey data suggested that most women believed that most of their family and friends (96%) and community (96%) will get the vaccine, again with high proportions of vaccinated, compared to unvaccinated, women reporting this (Table 2).

## Information

While changing social norms were shifting COVID-19 vaccine uptake and reducing hesitancy, 96% of respondents said they did not know where to get accurate information on COVID-19 and COVID-19 vaccines, and this proportion was similar among vaccinated and unvaccinated

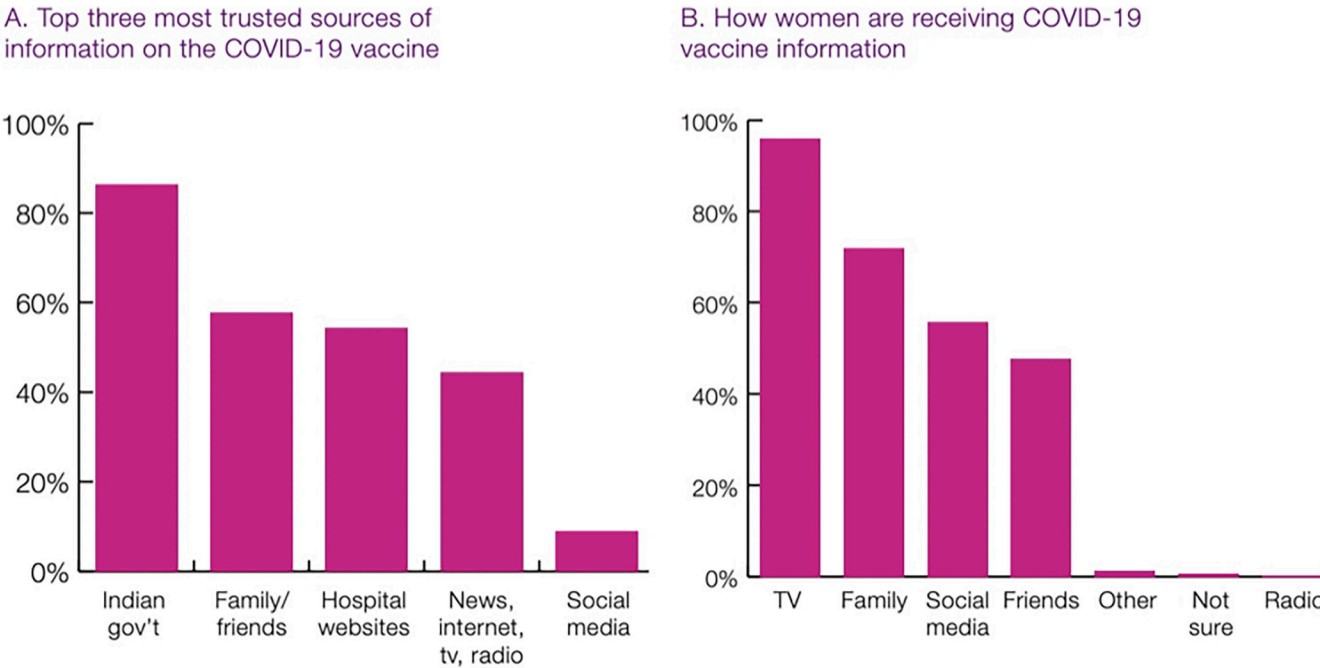

**Fig 2. Most trusted and actual sources of information about COVID-19 vaccine.**

women (Table 2). Women were keenly aware that they were seeing false and misleading information (61% said they saw information that they could not tell was true or false). And yet, there was a misalignment between who women said they trusted most for information, and where they got most of their information about COVID-19 vaccines (Fig 2). The most trusted sources were public health agencies/government, followed by family/friends, and then hospital websites (Fig 2). Social media was trusted by <10%. However, the main sources of information were TV (96%), followed by family (72%), social media (56%), and then friends (48%). This highlights that some people are getting their information from sources that they may not fully trust, potentially leading to more confusion or hesitancy.

There is a complex interplay between household decision-making, social norms, availability of reliable sources of information, and the deep trust in the government health care system that all must interact in the perfect balance to allow pregnant women in this setting to get vaccinated. One woman summed this up perfectly:

> "*All my family members have been vaccinated. Earlier somebody told us that pregnant ladies are not eligible for vaccination. But when I found the Prime Minister's statement regarding vaccination on google, then I told my husband about it. We discussed it with our gynecologist, and she told us that vaccination is good to take and then I had my vaccination. Earlier, my husband was worried about if the vaccination would harm the baby, but everything went well after vaccination. My husband says that everyone should take this vaccination if they are being asked to do so" (age 25,pregnant)*

### Factors associated with being vaccinated

In multi-variable logistics regression models including socio-demographics, the only factor associated with being vaccinated was whether the woman had a ration card, where women

**Table 3. Factors associated with being vaccinated among pregnant and postpartum women.**

| Variables | Model 1 | Model 2 | Model 3 | Model 4 | Model 5 | Model 6 |
|---|---|---|---|---|---|---|
| Currently Pregnant (compared to postpartum) | 1.08 (0.67–1.75) | 1.20 (0.73–1.97) | 1.20 (0.73–1.97) | 1.21 (0.73–2.03) | 1.06 (0.65–1.73) | 1.14 (0.69–1.86) |
| Age in years | 0.98 (0.92–1.05) | 0.98 (0.91–1.05) | 0.98 (0.92–1.05) | 0.98 (0.91–1.06) | 0.98 (0.91–1.05) | 0.98 (0.91–1.05) |
| Secondary education (compared to <secondary) | 1.11 (0.67–1.85) | 1.09 (0.65–1.82) | 1.21 (0.72–2.03) | 1.03 (0.60–1.77) | 1.13 (0.67–1.89) | 1.09 (0.65–1.83) |
| Hindu (compared to all others) | 0.65 (0.26–1.64) | 0.66 (0.26–1.68) | 0.67 (0.26–1.72) | 0.60 (0.23–1.59) | 0.58 (0.22–1.47) | 0.74 (0.29–1.90) |
| General caste (compared to all others) | 0.79 (0.46–1.35) | 0.74 (0.43–1.28) | 0.75 (0.43–1.29) | 0.76 (0.43–1.32) | 0.77 (0.45–1.33) | 0.76 (0.44–1.32) |
| Has ration card | 1.85** (1.12–3.04) | 1.89** (1.14–3.14) | 1.92** (1.15–3.20) | 1.86** (1.10–3.16) | 1.92** (1.16–3.18) | 1.86** (1.11–3.09) |
| Vaccine is effective | | 1.46 (0.36–6.02) | | | | |
| Vaccine is safe for baby | | 2.67* (0.97–7.39) | | | | |
| High trust in public health agencies | | | 1.53 (0.42–5.60) | | | |
| High trust in perinatal care provider | | | 2.93 (0.79–10.81) | | | |
| Perinatal provider recommended to get vaccine | | | | | | |
| Perinatal provider recommended NOT to get vaccine | | | | 0.19*** (0.09–0.37) | | |
| Perinatal provider did not mention vaccine | | | | 0.56* (0.31–1.02) | | |
| Heard information that couldn't tell if was true or false | | | | | 1.91** (1.14–3.18) | |
| Do you think your family and friends will/have gotten the vaccine? | | | | | | 1.51 (0.13–17.46) |
| Constant | 0.74 (0.11–4.79) | 0.22 (0.02–2.20) | 0.16 (0.02–1.49) | 1.64 (0.22–12.36) | 0.54 (0.08–3.70) | 0.52 (0.02–11.50) |
| Observations | 300 | 300 | 300 | 300 | 300 | 287 |

ciEform in parentheses

*** p<0.01

** p<0.05

* p<0.1

with a ration card (an indicator of poverty status) had increased odds of being vaccinated (OR = 1.85, 95% CI = 1.12–3.04) (Table 3). In the next set of models on belief in the vaccine, believing that the vaccine was safe for the baby was associated with increased odds of being vaccinated with marginal significance (p = 0.058) (OR = 2.67, 95% CI = 0.97–7.39). Neither of the variables related to trust in providers was significantly associated, however, a woman having her perinatal provider tell her not to get the vaccine (OR = 0.19, 95% CI = 0.09–0.37) and not discuss it (0.56, 95% CI = 0.31–1.02, p = 0.058) were both associated with lower odds of being vaccinated. A woman saying that she heard or saw information about the vaccine that she could not tell if was true or false was associated with increased odds of vaccination (OR = 1.91, 95% CI = 1.14–3.18). Belief that her family would all get vaccinated was not significantly associated.

## Discussion

Our research with pregnant and breastfeeding women indicated that some women and their communities were hesitant to receive a COVID-19 vaccine, especially initially. Vaccine hesitancy is complex [20] and misinformation and disinformation are often highlighted as the key influencers driving mistrust in the COVID-19 vaccine [21]. However, our finding highlighted that as more of the population has been safely vaccinated, unvaccinated individuals have

become more willing to do so themselves. Believing that the vaccine is safe, especially for the infant, is essential, as came out in the qualitative and quantitative results of this study. Addressing misinformation about side effects or impacts of the vaccine is clearly key—as these myths were pervasive among respondents and potentially driving some of the barriers enforced and lack of action of health care providers.

However, aside from this, healthcare provider refusal to vaccinate was a primary contributor to women in our sample being unable to receive a COVID-19 vaccine. This finding came out strongly in the qualitative and quantitative data—women wanted the vaccine while pregnant but were being advised not to get it, or were outright refused the vaccine. It is clear that education to providers about the safety of the vaccine in pregnancy was a missed opportunity and led to many women remaining unvaccinated in pregnancy, despite a desire for the contrary. Education would need to target all levels of health care providers, from specialized perinatal care providers to community health workers, and likely be ongoing and multi-pronged. Additionally, on a broader level, helping providers think through balancing the reality that pregnancy is a time of higher risk in some ways with the fact that COVID-19 in pregnancy posed many risks could have helped shift the paradigm.

One of the strongest predictors of not being vaccinated was a woman's perinatal provider either recommending against it, but also the provider simply not bringing it up. Trust in health care providers, including community health workers, was very high, as has been found in past studies in India [22]. A study from India among pregnant women and their health care providers regarding influenza vaccinations also found that health care providers were the most trusted sources for advice on the vaccination, and also that despite high interest among women, there was more uncertainty among their health care providers—although it did not seem as extreme as in this case [23]. Perhaps the novelty of the COVID-19 vaccine made providers additionally hesitant; the same could be true for hesitancy among community members as well, who are used to and accepting of other vaccines in pregnancy, such as tetanus. While we might expect that women would trust their providers and not get vaccinated if their provided recommended against it, its notable that even a provider not mentioning the vaccine contributed to women not getting vaccinated. This highlights the important role that silence can play, and the need to encourage providers to play an active role in promoting the vaccine in this population.

Much focus in current research on vaccine hesitancy has highlighted the role of misinformation (false information) and it was interesting to note that a majority of women in our sample reported that they saw information that they could not tell if was true or false. While it is heartening that so many women were aware and questioning the information they saw, it is also interesting that women reporting seeing questionable information was actually associated with vaccination status. This suggests that perhaps interventions that help people think about the veracity of the information that they see or even promoting questioning could be important to study further for addressing vaccine hesitancy.

Interestingly, we find that women with a ration card, which indicates that a household is eligible for subsidized foods (a marker of poverty), were more likely to be vaccinated. Caste, religion, and education, which are also often associated with health inequalities, were not associated with vaccination status in this analysis. A study among the general population (not focused on pregnant women) in India found that religion and education were not associated with vaccine acceptance, however, though they collected data on ration cards, they did not include this in their models [24]. It is possible that people with ration cards were more likely to be connected to the health system, perhaps even targeted by providers for vaccinations, or more likely to be accepting of government programs (such as vaccination).

In order to ensure that pregnant and breastfeeding women who want the COVID-19 vaccine can receive it, we need a multi-pronged gender and health-behavior informed strategy.

First, we suggest that government and civil society should invest resources in educating female, front-line healthcare workers (who provide the majority of care to pregnant and breastfeeding women), to ensure they are well informed about the benefits of vaccinating pregnant and breastfeeding women against COVID-19. Ensuring that female providers are strong advocates for vaccinating pregnant women against COVID-19 is likely to have significant benefits, given high levels of trust in the government health care system by the general public. Second, we recommend the development of widespread public health education campaigns about the COVID-19 vaccine that specifically target women to ensure they are educated about COVID-19 vaccine safety, with a focus on safety in pregnancy and for the infant. Women in this setting often face restrictions on their mobility or empowerment, and thus educating women alone is insufficient—messaging must also target and engage family members both to be supportive of COVID-19 vaccination in pregnancy, but also more broadly to address gender equality. Third, as the pandemic continues, it is critical that we continue to conduct high quality research into underlying causes of vaccine refusal or postponement in this high-risk population in India.

While this study adds to the literature by studying vaccine hesitancy and uptake among an often-neglected population who is of high risk (pregnant and postpartum women in rural India), this study does have limitations. Data is cross sectional, and thus while we look at associations we are unable to make claims about directionality of associations. Participants were recruited from one part of India, and thus findings might not be generalizable to other parts of India or outside of India. Respondents were also recruited through a community organization that provides services to pregnant and postpartum women, and thus, these respondents might be more connected to the health care system and have had more access to information about and actual uptake of COVID-19 vaccines. Additionally, all women had phones, and thus, while a large proportion noted having ration cards (a marker of poverty), this sample did not represent the least well-off populations in this setting (those without access to phones). Collecting data over the phone also posed limitations including (a) challenges in finding a suitable time for interviews and potential lack of privacy for the respondents (b) clarity of communication due to echo and voice cracking and (c) having to break up interviews over multiple phone calls for some participants.

## Conclusions

The results of this study surprised us and made us reconsider some of our own assumptions about vaccine uptake and barriers. The findings also made us question the course that we, as a public health community, have been taking to address low vaccine uptake. Many pregnant women in northern India who want the COVID-19 vaccine are not receiving their COVID-19 vaccines, despite it being approved in this population for over 6 months and pregnant women being high risk for adverse maternal and fetal impacts from COVID-19. To reach the underserved and high risk population of pregnant and breastfeeding women, it is imperative that India's COVID-19 vaccine outreach simultaneously target households [25], health care workers who are frontline along with other health care providers, and address social norms to increase vaccine uptake. Considering the complex interplay between trust in the government and health care providers, gender and social dynamics and norms, and other factors aside from supply are key to understanding and address vaccine hesitancy and vaccination—for COVID-10 and likely other vaccinations now and in the future, in India and globally.

## Supporting information

**S1 Text. Inclusivity in global research.**
(DOCX)

## Author Contributions

**Conceptualization:** Nadia G. Diamond-Smith.

**Data curation:** Nadia G. Diamond-Smith, Navneet Gill, Jagriti Gupta, Jasmeet Kaur.

**Formal analysis:** Nadia G. Diamond-Smith, Preetika Sharma, Mona Duggal, Navneet Gill, Katy Bradford Vosburg, Alison M. El Ayadi.

**Funding acquisition:** Nadia G. Diamond-Smith, Alison M. El Ayadi.

**Investigation:** Preetika Sharma, Mona Duggal, Vijay Kumar, Jasmeet Kaur, Pushpendra Singh, Katy Bradford Vosburg, Alison M. El Ayadi.

**Methodology:** Mona Duggal, Pushpendra Singh, Alison M. El Ayadi.

**Project administration:** Navneet Gill, Jagriti Gupta, Vijay Kumar, Pushpendra Singh, Katy Bradford Vosburg, Alison M. El Ayadi.

**Resources:** Vijay Kumar, Katy Bradford Vosburg.

**Software:** Pushpendra Singh.

**Supervision:** Pushpendra Singh, Alison M. El Ayadi.

**Validation:** Vijay Kumar, Jasmeet Kaur.

**Visualization:** Katy Bradford Vosburg.

**Writing – original draft:** Nadia G. Diamond-Smith, Katy Bradford Vosburg, Alison M. El Ayadi.

**Writing – review & editing:** Mona Duggal, Vijay Kumar, Jasmeet Kaur, Pushpendra Singh.

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
