## [Decision Letter · Decision Letter 0]

21 Jun 2022

PGPH-D-22-00468

The supply is there. So why can’t pregnant and breastfeeding women in rural India get the COVID-19 vaccine?

Dear Dr. Diamond-Smith,

Thank you for submitting your manuscript to PLOS Global Public Health. After careful consideration, we feel that it has merit but does not fully meet PLOS Global Public Health’s publication criteria as it currently stands. Therefore, we invite you to submit a revised version of the manuscript that addresses the points raised during the review process.

This study is an important one, but currently the manuscript has has too little information to engage a reader ( for instance gaps in description of the context/setting, no socio-demographic tables or sample descriptions, no descriptive tables from the survey and no segregated themes from the qualitative data to name a few). One of the reviewers has pointed out important gaps which need to be addressed before due consideration on its merit for publication.  

Please submit your revised manuscript by . If you will need more time than this to complete your revisions, please reply to this message or contact the journal office at globalpubhealth@plos.org. Please include the following items when submitting your revised manuscript:

We look forward to receiving your revised manuscript.

Kind regards,

Prashanth Nuggehalli Srinivas, MBBS, MPH, PhD

Academic Editor

Journal Requirements:

a. Please clarify all sources of funding (financial or material support) for your study. List the grants (with grant number) or organizations (with url) that supported your study, including funding received from your institution.

b. State the initials, alongside each funding source, of each author to receive each grant.

c. State what role the funders took in the study. If the funders had no role in your study, please state: “The funders had no role in study design, data collection and analysis, decision to publish, or preparation of the manuscript.”

3. Please ensure that the funders and grant numbers match between the Financial Disclosure field and the Funding Information tab in your submission form. Note that the funders must be provided in the same order in both places as well.

4. Please update your online Competing Interests statement. If you have no competing interests to declare, please state: “The authors have declared that no competing interests exist.”

5. In the online submission form, you indicated that “The data underlying this article will be shared on reasonable request to the corresponding author.”. All PLOS journals now require all data underlying the findings described in their manuscript to be freely available to other researchers, either 1. In a public repository, 2. Within the manuscript itself, or 3. Uploaded as supplementary information.

Additional Editor Comments (if provided):

Reviewers' comments:

Reviewer's Responses to Questions

**Comments to the Author**

1. Does this manuscript meet PLOS Global Public Health’s publication criteria? Is the manuscript technically sound, and do the data support the conclusions? The manuscript must describe methodologically and ethically rigorous research with conclusions that are appropriately drawn based on the data presented.

Reviewer #1: Partly

Reviewer #2: Yes

2. Has the statistical analysis been performed appropriately and rigorously?

Reviewer #1: No

Reviewer #2: I don't know

3. Have the authors made all data underlying the findings in their manuscript fully available (please refer to the Data Availability Statement at the start of the manuscript PDF file)?

Reviewer #1: No

Reviewer #2: Yes

4. Is the manuscript presented in an intelligible fashion and written in standard English?

Reviewer #1: Yes

Reviewer #2: Yes

5. Review Comments to the Author

Reviewer #1: Thank you for an opportunity to review this article, which attempts to address a very important topic of low uptake of the COVID-19 Vaccine among pregnant and lactating women in one rural setting in India. The authors seem to have made a rigorous attempt to capture data through mixed methods, but much of the details needed to interpret the paper are missing from the paper at present. I would suggest a revised paper with more contextual details, methodological nuances such as details of the sampling size and strategy, detailed reporting of quantitative tables (there are no tables) and qualitative findings (few scattered themes reported), and a deeper discussion section in relation to current literature.

More details are below

Introduction

Important dates, COVID- context in India particularly of vaccination, type of vaccines available, the CoWin App and digitalization -perhaps a timeline here? when did vaccination begin, when did it begin for pregnant women (highlighting the fact that it was not approved for pregnant women initially)

-a strong argument for why this study was done. What does it add to what we already know in literature?

- description of the context is important, currently there is no description of the study setting. If the state name/district was masked, perhaps how does this limit interpretations of the data. For instance, study (10) has shown that 93% of women reported intentions to get vaccinated. Was something different in this context?

Methods

Detailed description of survey methods- where was it done, sample size calculations, non response rates, survey tool and its dimensions ( might help to attach), data cleaning, analysis, etc

Details of qualitative methods- purposive sample, saturation, non-response, key themes that were asked, interview guide used for facilitation. Transcription, analyses

Details of the qualitative and quantitative demographics of participants needed.

Findings

It might make sense to separate the qualitative and quantitative findings and present them separately.

In the survey section of the findings, please report main data tables-example- intention to get vaccine,sources of information, trust, etc. also if the responses differed by age, family type, parity, the trimester they are in, other socio demographics. We don’t have much details of the 500 people surveyed in the findings. Discuss the key tables in details.

In the qualitative section, one can pull from the data table and try to explain some of the ‘whys’ of the findings. But this needs to be done in 3-5 well-defined themes, for right now there is a long write-up that is a bit difficult to read. For instance, lines 140 -152 have several themes jumbled up- reasons for not getting vaccinated, some idea of trust and on uptake, etc. Quotes need to be reported in lines with how qualitative data is usually reported (in double quotes, with some description of the participant demographics)- so that quotes can be interpreted by the reader with ease. More direct quotes can be included to give readers a better flavour of the field. One can also use tables to present summarized qualitative data or data from a few cases that highlight important issues. The richness of the interviews need to come out in this section

Discussion

-the limitations of the study need to be discussed. What were some of the limitations of doing phone interviews? How was non response dealt with? Do we leave out specific sections of the population ( the most vulnerable in fact) if we do phone interviews?

-lines 209-211 seem particularly important to me, based on what I know of the Indian context. But this point hasn’t been detailed in the findings section anywhere.

-what has been found from other contexts on pregnancy and vaccination. There is a lot of literature in the non-covid areas as well on this.

- What is particularly different about India. In fact in india, pregnant women are offered vaccines ( TT) regularly, what is it that made them hesitate with regard to this vaccine? Vaccination was also happening in the private sector- in this context, did the women have access to this option.

-line 220- if the main reason for being unable to get a covid vaccine was due to the provider hesitancy, then should we not be educating providers rather than women as being suggested here. perhaps one needs to look at the main 3-5 reasons for low uptake, and see what specific measures (long-term and short-term) can be taken to address these. A table here would help to summarize these measures.

Apologies again for the long list of suggestions. I feel that such topics are important to work with. It is just that the manuscript at present needs more work in order for it to convey the rigor of the methods used as well as the depths of the findings. My very best wishes for the revisions.

Reviewer #2: Dear authors,

This is an important, timely paper, and I commend you for conducting this research. I would like to make some suggestions that will tighten the arguments.

Introduction: 

- Lines 73-76: Be sure to specify that in a reversal of what was witnessed worldwide, where men had higher Covid-19 fatality rates than women, in India women had a higher fatality rate

- Lines 83-87: The first and second part of the sentence have little in common except temporality - perhaps flip the two parts

Methods:

- Please provide an overview of topics covered in the phone survey and in-depth interview questionnaire 

- What % of women from the phone survey agreed to a follow-up interview? What % of these women were eventually interviewed? Did those who agreed to be interviewed differ in any way from those who did not? (just on some basic demographic measures)

Findings:

- To what extent did the 27% who did not want the vaccination overlap with the 30% who worried about the safety of the mother/baby?

- What were the main concerns of the remaining 70% of mothers?

- One of the 'main reasons' for lack of vaccination was the unwillingness of healthcare providers - what % of women reported this? (I assume 75%, based on line 156, but am uncertain)

- With regard to trust, in line 144 it states that 90% of women trusted govt/public health agencies, and in line 155 it states that 90% trusted perinatal care providers - do these refer to the same statistic?

- Is there any data to back up the assertions about women's lack of mobility and empowerment?- Lines 173-176 seem out of place, perhaps move to the end of the findings?

- Please be a bit clear about which findings are from the survey and which from the interviews. 

- If possible, accompany quotes with some basic information about the woman (age, pregnancy/post-partum/nursing status)

- Fascinating finding - that trusted sources are at odds with sources women actually receive information from

Discussion: 

- Line 209: please clarify 'larger proportion'

- Although women's lack of mobility and empowerment is highlighted in the findings, it finds no place in the discussion (harder to address in an intervention, of course, but should be nevertheless be noted) 

- Similarly, while suggestions focus on targeting women, the findings suggest working with family members as well 

Overall, some of the statements and assertions made in the paper require some evidence to back them up.

6. PLOS authors have the option to publish the peer review history of their article (what does this mean?). If published, this will include your full peer review and any attached files.

**Do you want your identity to be public for this peer review?** For information about this choice, including consent withdrawal, please see our Privacy Policy.

Reviewer #1: No

Reviewer #2: No

---

## [Decision Letter · Decision Letter 1]

17 Oct 2022

PGPH-D-22-00468R1

The supply is there. So why can’t pregnant and breastfeeding women in rural India get the COVID-19 vaccine?

Dear Dr. Diamond-Smith,

Thank you for submitting your manuscript to PLOS Global Public Health. After careful consideration, we feel that it has merit but does not fully meet PLOS Global Public Health’s publication criteria as it currently stands. Therefore, we invite you to submit a revised version of the manuscript that addresses the points raised during the review process.

We look forward to receiving your revised manuscript.

Kind regards,

Prashanth Nuggehalli Srinivas, MBBS, MPH, PhD

Academic Editor

Journal Requirements:

Additional Editor Comments (if provided):

Reviewers' comments:

Reviewer's Responses to Questions

**Comments to the Author**

1. If the authors have adequately addressed your comments raised in a previous round of review and you feel that this manuscript is now acceptable for publication, you may indicate that here to bypass the “Comments to the Author” section, enter your conflict of interest statement in the “Confidential to Editor” section, and submit your "Accept" recommendation.

Reviewer #2: All comments have been addressed

2. Does this manuscript meet PLOS Global Public Health’s publication criteria? Is the manuscript technically sound, and do the data support the conclusions? The manuscript must describe methodologically and ethically rigorous research with conclusions that are appropriately drawn based on the data presented.

Reviewer #2: Yes

3. Has the statistical analysis been performed appropriately and rigorously?

Reviewer #2: Yes

4. Have the authors made all data underlying the findings in their manuscript fully available (please refer to the Data Availability Statement at the start of the manuscript PDF file)?

Reviewer #2: Yes

5. Is the manuscript presented in an intelligible fashion and written in standard English?

Reviewer #2: Yes

6. Review Comments to the Author

Reviewer #2: Dear authors,

Your effort in addressing comments that were raised with respect to the original submission is noted and appreciated. There is more clarity in the methodology, detail in the findings, and analysis in the discussion - and these revisions significantly strengthen the paper. But more remains to be done, described below in order of importance.

- In Table 2, is it possible to breakdown further between vaccinated and unvaccinated women? When not separated out, it is hard to identify specific drivers of vaccination/non-vaccination, and this is really the meat of the paper. Please distinguish between the two categories when describing your findings, and further reflect on them in the discussion.

- I understand the desire to protect participants, but by providing such little detail about the study site, a lot of context is lost. Would strongly suggest including some additional information.

There are many other points require additional clarification:

- Lines 129-130: are phrased ambiguously - were all participants from 2 districts, or 60% of them?

- Line 131: rank in the middle of what?

- Lines 129-134: were participants rural, or rural and suburban?

- Lines 135-142: were participants up to 6 months post-partum, or up to 1 year post-partum?

- Clarify how having or not having a ration card was a marker of poverty

- Note the version of Dedoose used

- Line 188: XX XX - I assume details will be filled in?

- Line 192: what is bot technology, and what is its relevance to this study?

- Lines 209-211: This an important point and needs to be moved up further

- Lines 211-213 - doesn't add up to 100% - what other categories included?

- Line 214+: Does this excerpt refer to one of the women in the 47%?

- Lines 224-225 show high trust in vaccines, but line 226 onwards shows some concerns - are these not at odds?

- Lines 244-245 - where is this stated in the hypothesis?

- Lines 274-275: "the health care system not wanting to vaccinate pregnant women" is awkward phrasing

- Line 320: In Factors associated with being vaccinated, p values are not provided consistently, so it is hard to know whether the finding is significant or not

- Lines 329-331 - is the association positive or negative?

- It is still not always clear what findings are from the phone surveys and/or the in-depth interviews

- Table 1: the religion/Hindu row is not filled

- The abstract is a little disorganized and I don't think it represents the content of paper effectively.

- Finally, the manuscript is filled with typos and grammatical errors: line 81 ("this is contrast"); line 105 ("vaccine uptake trust"); line 108 ("pregnancy women"); line 117 ("more, mixed methods"); line 141 ('oof' instead of 'of'); line 150 ("were called to again to..."); etc. I stopped noting these after one point, but please proof-read the paper carefully.

I do believe these are minor revisions and can be addressed easily. This is an important paper and it deserves to be published, and I look forward to seeing it in print.

Regards.

7. PLOS authors have the option to publish the peer review history of their article (what does this mean?). If published, this will include your full peer review and any attached files.

**Do you want your identity to be public for this peer review?** For information about this choice, including consent withdrawal, please see our Privacy Policy.

Reviewer #2: No

---

## [Editor Report · Decision Letter 2]

4 Nov 2022

The supply is there. So why can’t pregnant and breastfeeding women in rural India get the COVID-19 vaccine?

PGPH-D-22-00468R2

Dear Dr. Diamond-Smith,

We are pleased to inform you that your manuscript 'The supply is there. So why can’t pregnant and breastfeeding women in rural India get the COVID-19 vaccine?' has been provisionally accepted for publication in PLOS Global Public Health.

Best regards,

Prashanth Nuggehalli Srinivas, MBBS, MPH, PhD

Academic Editor